# Methods of Moment and Maximum Entropy for Solving Nonlinear Expectation

**Lei Gao** *  **and Dong Han**

Department of Mathematics, Shanghai Jiao Tong University, Shanghai 200240, China; donghan@sjtu.edu.cn
* Correspondence: sunnydoll@alumni.sjtu.edu.cn

**Abstract:** In this paper, we consider a special nonlinear expectation problem on the special parameter space and give a necessary and sufficient condition for the existence of the solution. Meanwhile, we generalize the necessary and sufficient condition to the two-dimensional moment problem. Moreover, we use the maximum entropy method to carry out a kind of concrete solution and analyze the convergence for the maximum entropy solution. Numerical experiments are presented to compute the maximum entropy density functions.

**Keywords:** moment problem; maximum entropy; nonlinear expectation; existence and uniqueness

## 1. Introduction

The sublinear expectation $\hat{\mathbb{E}}$ introduced by Peng [1,2] can be regarded as the supremum of a family of linear expectations $\{E_{\boldsymbol{\theta}} : \boldsymbol{\theta} \in \Theta\}$, that is,

$$\hat{\mathbb{E}}[\varphi(X)] = \sup_{\boldsymbol{\theta} \in \Theta} E_{\boldsymbol{\theta}}[\varphi(X)], \tag{1}$$

where $\varphi(x)$ is a local Lipschitz continuous function and $\Theta$ is the parameter space.

It is evident that the sublinear expectation defined by (1) depends on the choice of parameter space $\Theta$. Different spaces will result in different nonlinear expectations. In particular, let $\varphi(x) = x^n$ and:

$$\bar{\mu}_n = \hat{\mathbb{E}}[X^n] \geq \underline{\mu}_n = -\hat{\mathbb{E}}[-X^n], \quad n = 1, 2, \cdots, N, \tag{2}$$

then the parameter space $\Theta$ can be chosen as the following form:

$$\Theta = [\underline{\mu}_1, \bar{\mu}_1] \times [\underline{\mu}_2, \bar{\mu}_2] \times \cdots \times [\underline{\mu}_N, \bar{\mu}_N]. \tag{3}$$

When $N = 1$, Peng [3] gave the definition of the independent and identically distributed random variable and proved the weak law of large numbers (LLN) under the sublinear expectation and Condition (2). Furthermore, if $\Theta = \{0\} \times [\underline{\mu}_2, \bar{\mu}_2]$, then Peng [4–8] defined the G-normal distribution and presented a new central limit theorem (CLT) under $\hat{\mathbb{E}}$. The new LLN and CLT are the theoretical foundations in the framework of sublinear expectation.

The calculation of $\hat{\mathbb{E}}(\varphi(X))$ can be performed by solving the following nonlinear partial differential equation:

$$\begin{cases} \dfrac{\partial u}{\partial t} - \dfrac{1}{2}\left(\left(\dfrac{\partial^2 u}{\partial x^2}\right)^+ - \sigma^2\left(\dfrac{\partial^2 u}{\partial x^2}\right)^-\right) = 0, & (t,x) \in [0,+\infty) \times \mathbb{R}, \\ u(0,x) = \varphi(x), & x \in \mathbb{R}, \end{cases} \tag{4}$$

whose solution is $u(t,x) = \hat{\mathbb{E}}[\varphi(x + \sqrt{t}X)]$. When the initial value $\varphi(x)$ is a convex function, Peng [8] gave the expression of $\hat{\mathbb{E}}[\varphi(X)]$ as follows:

$$\hat{\mathbb{E}}[\varphi(X)] = \frac{1}{\sqrt{2\pi\bar{\mu}_2^2}} \int_{-\infty}^{+\infty} \varphi(x)\exp\left\{-\frac{x^2}{2\bar{\mu}_2^2}\right\}dx. \tag{5}$$

If $\varphi(x)$ is a concave function, the variance $\bar{\mu}_2^2$ in (5) will be replaced by $\underline{\mu}_2^2$. For neither the concave nor the convex case, Hu [9] derived the explicit solutions of Problem (4) with the initial condition $\varphi(x) = x^n$, $n \geq 1$. Gong [10] used the fully-implicit numerical scheme to compute the nonlinear probability under the G-expectation determined by the G-heat Equation (4) with the initial condition $\varphi(x) = \mathbf{1}_{\{x<0\}}$, $x \in \mathbb{R}$. Here, $\mathbf{1}_{\{x<0\}}$ denotes the indicator function of the set $\{x|x < 0\}$. What all these methods mentioned above have in common is that G-expectation is calculated via solving the nonlinear partial differential Equation (4) with the particular initial condition $\varphi(x)$. However, because of the nonlinear term, it is not easy to find a solution of (4) with general continuous initial function $\varphi(x)$.

Based on the above reasons, in this paper, we consider a kind of special parameter space $\Theta$ defined in (3) and convert the sublinear expectation in (1) into the following two series of moment problems: find the probability density functions $\bar{p}(x)$ and $\underline{p}(x)$ such that:

$$\bar{\mu}_n = \int_{-\infty}^{+\infty} x^n \bar{p}(x)dx \tag{6}$$

and:

$$\underline{\mu}_n = \int_{-\infty}^{+\infty} x^n \underline{p}(x)dx, \tag{7}$$

respectively, where $n = 0, 1, \cdots, N$ and $N = 2M$. That is, we will approximately find a special class of nonlinear expectations $\hat{\mathbb{E}}$, which satisfies:

$$\hat{\mathbb{E}}[X^n] = \sup_{\boldsymbol{\theta} \in \Theta} \int_{\mathbb{R}} x^n p_{\boldsymbol{\theta}}(x)dx = \sup_{\underline{\mu}_n \leq \theta_n \leq \bar{\mu}_n} \int_{\mathbb{R}} x^n p_{\boldsymbol{\theta}}(x)dx = \bar{\mu}_n$$

and:

$$-\hat{\mathbb{E}}[-X^n] = -\sup_{\boldsymbol{\theta} \in \Theta} \int_{\mathbb{R}} (-x^n) p_{\boldsymbol{\theta}}(x)dx = -\inf_{\underline{\mu}_n \leq \theta_n \leq \bar{\mu}_n} \int_{\mathbb{R}} x^n p_{\boldsymbol{\theta}}(x)dx = \underline{\mu}_n,$$

where $\theta = (\theta_0, \theta_1, \cdots, \theta_N)$.

The rest of this article is organized as follows. In Section 2, we present an alternative sufficient and necessary condition of the existence of solutions $\bar{p}(x)$ and $\underline{p}(x)$ that satisfy (6) and (7), respectively. In Section 3, we use the maximum entropy method to find the concrete solutions and analyze the convergence of maximum entropy solutions. In Section 4, we conduct the numerical simulations to calculate the maximum entropy density functions.

## 2. Existence of Solutions for Moment Problems

According to Theorems 1.35 and 1.36 in Akihito [11], the sequences $\{\bar{\mu}_j\}_{j=0}^{N}$ and $\{\underline{\mu}_j\}_{j=0}^{N}$ should satisfy some conditions if we use them to determine the probability density functions. Therefore, in this

section, we consider the sufficient and necessary conditions to the existence of solutions for the moment Problems (6) and (7), respectively.

Let:

$$\bar{\Delta}_{2M} = \begin{pmatrix} \bar{\mu}_0 & \bar{\mu}_1 & \cdots & \bar{\mu}_M \\ \bar{\mu}_1 & \bar{\mu}_2 & \cdots & \bar{\mu}_{M+1} \\ \vdots & \vdots & & \vdots \\ \bar{\mu}_M & \bar{\mu}_{M+1} & \cdots & \bar{\mu}_{2M} \end{pmatrix} \text{ and } \underline{\Delta}_{2M} = \begin{pmatrix} \underline{\mu}_0 & \underline{\mu}_1 & \cdots & \underline{\mu}_M \\ \underline{\mu}_1 & \underline{\mu}_2 & \cdots & \underline{\mu}_{M+1} \\ \vdots & \vdots & & \vdots \\ \underline{\mu}_M & \underline{\mu}_{M+1} & \cdots & \underline{\mu}_{2M} \end{pmatrix} \tag{8}$$

be the Hankel matrices of the given sequences $\{\bar{\mu}_j\}_{j=0}^{2M}$ and $\{\underline{\mu}_j\}_{j=0}^{2M}$, respectively.

The following theorem gives an alternative sufficient and necessary condition for the existence of $\bar{p}(x)$, which satisfies (6).

**Theorem 1.** *A sufficient and necessary condition that the sequence $\{\bar{\mu}_j\}_{j=0}^{N}$ determines the probability density function $\bar{p}(x)$ is that $\{\bar{\mu}_j\}_{j=0}^{N}$ satisfies one of the following two conditions:*
*(a) For any nonvanishing vector $\mathbf{X}_m^T = (x_0, x_1, \cdots, x_m)$ $(1 \le m \le M)$,*

$$\bar{\mu}_{2m} \mathbf{X}_{m-1}^T \bar{\Delta}_{2(m-1)} \mathbf{X}_{m-1} > (\bar{U}_m^T \mathbf{X}_{m-1})^2;$$

*(b) If there exist $m_0 \ge 1$ and nonvanishing vector $\mathbf{X}_{m_0}^T = (x_0, x_1, \cdots, x_{m_0})$ such that $\mathbf{X}_{m_0}^T \bar{\Delta}_{2m_0} \mathbf{X}_{m_0} = 0$, then:*

$$|\bar{\Delta}_{2m}| = 0$$

*for $m_0 \le m \le M$, where $\bar{U}_m^T = (\bar{\mu}_m, \cdots, \bar{\mu}_{2m-1})$.*

**Proof.** We prove first the necessity. Let $X$ be a continuous random variable with density function $\bar{p}(x)$, whose original moments are:

$$\bar{\mu}_m = E[X^m] = \int_R x^m \bar{p}(x)dx, \quad m = 0, 1, \cdots, 2M, \tag{9}$$

where $\bar{\mu}_0 = 1$.

For any $m + 1$-dimensional nonvanishing vector $\mathbf{X}_m = (x_0, x_1, \cdots, x_m)^T$, we can check that:

$$\mathbf{X}_m^T \bar{\Delta}_{2m} \mathbf{X}_m = E[(\sum_{j=0}^{m} x_j X^j)^2], \quad m = 0, 1, \cdots, M. \tag{10}$$

In fact, taking $m = 0$, we have:

$$\mathbf{X}_0^T \bar{\Delta}_0 \mathbf{X}_0 = x_0^2 = E[x_0^2], \tag{11}$$

which means that (10) holds for $m = 0$.

Assume that the Equation (10) holds for $0 \le m \le M - 1$, that is,

$$\mathbf{X}_{M-1}^T \bar{\Delta}_{2(M-1)} \mathbf{X}_{M-1} = E[(\sum_{j=0}^{M-1} x_j X^j)^2]. \tag{12}$$

Then, we consider the case of $m = M$ and note that the matrix $\bar{\Delta}_{2M}$ can be rewritten as:

$$\bar{\Delta}_{2M} = \begin{pmatrix} \bar{\Delta}_{2(M-1)} & \bar{U}_M \\ \bar{U}_M^T & \bar{\mu}_{2M} \end{pmatrix}. \tag{13}$$

Multiply (13) by $\mathbf{X}_M^T$ and $\mathbf{X}_M$, respectively; it follows from (9) and (12) that:

$$
\begin{aligned}
\mathbf{X}_M^T \bar{\Delta}_{2M} \mathbf{X}_M &= (\mathbf{X}_{M-1}^T, x_M) \begin{pmatrix} \bar{\Delta}_{2(M-1)} & \bar{U}_M \\ \bar{U}_M^T & \bar{\mu}_{2M} \end{pmatrix} \begin{pmatrix} \mathbf{X}_{M-1} \\ x_M \end{pmatrix} \\
&= \mathbf{X}_{M-1}^T \bar{\Delta}_{2(M-1)} \mathbf{X}_{M-1} + 2x_M \bar{U}_M^T X_{M-1} + \bar{\mu}_{2M} x_M^2 \\
&= E[(\sum_{j=0}^{M-1} x_j X^j)^2] + 2x_M \sum_{j=0}^{M-1} \bar{\mu}_{M+j} x_j + \bar{\mu}_{2M} x_M^2 \\
&= E[(\sum_{j=0}^{M-1} x_j X^j)^2] + 2x_M \sum_{j=0}^{M-1} x_j E[X^{M+j}] + x_M^2 E[X^{2M}] \\
&= E[(\sum_{j=0}^{M} x_j X^j)^2] \ge 0.
\end{aligned}
\tag{14}
$$

By using mathematical induction, we know that (10) holds for $m = 0, 1, \cdots, M$. Moreover, Equation (10) means that the Hankel matrices are nonnegative definite.

Now, we prove the necessary condition. Because of the nonnegative definiteness of $\bar{\Delta}_{2m}$ ($0 \le m \le M$), there are two possible cases for $\mathbf{X}_m^T \bar{\Delta}_{2m} \mathbf{X}_m$, that is $\mathbf{X}_m^T \bar{\Delta}_{2m} \mathbf{X}_m > 0$ for any $m$-dimensional nonzero vector $\mathbf{X}_m$ or $\mathbf{X}_{m_0}^T \bar{\Delta}_{2m_0} \mathbf{X}_{m_0} = 0$ for some $m_0$-dimensional nonzero vector $\mathbf{X}_{m_0}$. Thus, we divide the proof into two cases.

Case 1: Let $\mathbf{X}_m^T \bar{\Delta}_{2m} \mathbf{X}_m = E[(\sum_{j=0}^{m} x_j X^j)^2] > 0$ for any nonzero vector $\mathbf{X}_m$ ($1 \le m \le M$); by (14), we have:

$$E[(\sum_{j=0}^{m} x_j X^j)^2] = \mathbf{X}_{m-1}^T \bar{\Delta}_{2(m-1)} \mathbf{X}_{m-1} + 2x_m \bar{U}_m^T X_{m-1} + \bar{\mu}_{2m} x_m^2. \tag{15}$$

We regard (15) as a quadratic equation in one variable with respect to $x_m$. Since the value of (15) is greater than zero, so its discriminant is less than zero, i.e.,

$$(\bar{U}_m^T \mathbf{X}_{m-1})^2 - \bar{\mu}_{2m} \mathbf{X}_{m-1}^T \bar{\Delta}_{2(m-1)} \mathbf{X}_{m-1} < 0. \tag{16}$$

This implies that (a) in Theorem 1 holds for $m = 1, 2, \cdots, M$.

Case 2: Let $\mathbf{X}_{m_0-1}^T \bar{\Delta}_{2(m_0-1)} \mathbf{X}_{m_0-1} = 0$ for some nonvanishing vector $\mathbf{X}_{m_0-1}$ ($2 \le m_0 \le M+1$). For $m > m_0 - 1$, let $m = m_0 - 1 + k_0$, $k_0 = 1, 2, \cdots, M - m_0 + 1$. We choose an $m \times 1$ vector $\mathbf{X}_m^T = (\mathbf{X}_{m_0-1}^T, \underbrace{0, \cdots, 0}_{k_0})$ and multiply $\bar{\Delta}_{2m}$ by $X_m^T$ and $X_m$. Here, we consider only the case of $k_0 = 1$ as an example, and the same method can be used to analyze for $2 \le k_0 \le M - m_0 + 1$. Then,

$$
\begin{aligned}
\mathbf{X}_{m_0}^T \bar{\Delta}_{2m_0} \mathbf{X}_{m_0} &= (\mathbf{X}_{m_0-1}^T, 0) \begin{pmatrix} \bar{\Delta}_{2(m_0-1)} & \bar{U}_{m_0} \\ \bar{U}_{m_0}^T & \bar{\mu}_{2m_0} \end{pmatrix} \begin{pmatrix} \mathbf{X}_{m_0-1} \\ 0 \end{pmatrix} \\
&= \mathbf{X}_{m_0-1}^T \bar{\Delta}_{2(m_0-1)} \mathbf{X}_{m_0-1}.
\end{aligned}
\tag{17}
$$

Since $\bar{\Delta}_{2m_0}$ is nonnegative definite and $\mathbf{X}_{m_0-1}^T \bar{\Delta}_{2(m_0-1)} \mathbf{X}_{m_0-1} = 0$, by (17), it follows that $|\bar{\Delta}_{2m_0}| = 0$, that is (b) holds.

Next, we prove the sufficiency via its converse-negative proposition. In other words, we need to show that there is no probability density function $p(x)$ such that the corresponding moments $\{\bar{\mu}_j\}_{j=0}^N$ satisfy: there exist a positive integer $m_0$ $(2 \le m_0 \le M+1)$ and an $m_0$-dimensional nonzero vector $\mathbf{X}_{m_0-1}$ such that:

(i) $\bar{\mu}_{2m_0} \mathbf{X}_{m_0-1}^T \bar{\Delta}_{2(m_0-1)} \mathbf{X}_{m_0-1} \le (\bar{U}_{m_0}^T \mathbf{X}_{m_0-1})^2$;

(ii) if $\mathbf{X}_{m_0-1}^T \bar{\Delta}_{2(m_0-1)} \mathbf{X}_{m_0-1} = 0$, then there exists a positive integer $m_1 > m_0 - 1$ such that $|\bar{\Delta}_{2m_1}| > 0$.

Now, we carry out the proof by contradiction. Suppose such a density function $\bar{p}(x)$ exists. If $\mathbf{X}_{m_0}^T \bar{\Delta}_{2m_0} \mathbf{X}_{m_0} > 0$ holds for any $m_0 + 1$-dimensional nonvanishing vector, then it follows from (15) that:

$$(\bar{U}_{m_0}^T \mathbf{X}_{m_0-1})^2 - \bar{\mu}_{2m_0} \mathbf{X}_{m_0-1}^T \bar{\Delta}_{2(m_0-1)} \mathbf{X}_{m_0-1} < 0. \tag{18}$$

The above inequality contradicts Condition (i).

If the moments $\bar{\mu}_j$ satisfy Condition (ii), then there exists an $m_0$-dimensional nonzero vector such that $\mathbf{X}_{m_0-1}^T \bar{\Delta}_{2(m_0-1)} \mathbf{X}_{m_0-1} = 0$. By Condition (ii), without loss of generality, we let $m_1 = m_0$ and repeat the same procedure for the derivation of (17) to get that:

$$0 < \mathbf{X}_{m_0}^T \bar{\Delta}_{2m_0} \mathbf{X}_{m_0} = \mathbf{X}_{m_0-1}^T \bar{\Delta}_{2(m_0-1)} \mathbf{X}_{m_0-1} = 0, \tag{19}$$

where $\mathbf{X}_{m_0}^T = (\mathbf{X}_{m_0-1}^T, 0)$. Obviously, the relationship in (19) is contradictory.

Then, we complete the proof of necessity. □

**Remark 1.** *We use $\underline{\mu}_m$, $\underline{\Delta}_{2m}$, and $\underline{U}_m$ to replace $\bar{\mu}_{2m}$, $\bar{\Delta}_{2m}$, and $\bar{U}_m$, respectively, in Theorem 1, then we get the sufficient and necessary conditions to the existence of the probability density function $\underline{p}(x)$.*

Note that Theorem 1 is presented in one-dimensional random variable space. Now, we extend this theorem to two-dimensional case. Let $X$ and $Y$ be any continuous random variables with the joint probability density function $p_\theta(x, y)$. The two-dimensional moment problems are defined by: find the joint probability density function $\bar{p}(x, y)$ and $\underline{p}(x, y)$ such that:

$$\bar{\mu}_{ij} = \int_{\mathbb{R}} \int_{\mathbb{R}} x^i y^j \bar{p}(x, y) dx dy, \quad i, j \in \mathcal{M} \tag{20}$$

and:

$$\underline{\mu}_{ij} = \int_{\mathbb{R}} \int_{\mathbb{R}} x^i y^j \underline{p}(x, y) dx dy, \quad i, j \in \mathcal{M}, \tag{21}$$

where the set:

$$\mathcal{M} = \{(i, j) | i, j = 0, 1, \cdots, 2M \text{ and } 0 \le i + j \le 2M\}.$$

Here, we still take the sequence $\{\bar{\mu}_{ij}\}_{i,j \in \mathcal{M}}$ for example and present the main results in the following theorem without proof. Let:

$$
\bar{\Gamma}_{2m+1} := \begin{pmatrix}
\bar{\mu}_{00} & \bar{\mu}_{10} & \cdots & \bar{\mu}_{m0} & \bar{\mu}_{01} & \bar{\mu}_{02} & \cdots & \bar{\mu}_{0m} \\
\bar{\mu}_{10} & \bar{\mu}_{20} & \cdots & \bar{\mu}_{m+1,0} & \bar{\mu}_{11} & \bar{\mu}_{12} & \cdots & \bar{\mu}_{1m} \\
\vdots & \vdots & & \vdots & \vdots & \vdots & & \vdots \\
\bar{\mu}_{m0} & \bar{\mu}_{m+1,0} & \cdots & \bar{\mu}_{2m,0} & \bar{\mu}_{m1} & \bar{\mu}_{m2} & \cdots & \bar{\mu}_{mm} \\
\bar{\mu}_{01} & \bar{\mu}_{11} & \cdots & \bar{\mu}_{m1} & \bar{\mu}_{02} & \bar{\mu}_{02} & \cdots & \bar{\mu}_{0,m+1} \\
\vdots & \vdots & & \vdots & \vdots & \vdots & & \vdots \\
\bar{\mu}_{0m} & \bar{\mu}_{1m} & \cdots & \bar{\mu}_{mm} & \bar{\mu}_{0,m+1} & \bar{\mu}_{0,m+2} & \cdots & \bar{\mu}_{0,2m}
\end{pmatrix} \tag{22}
$$

be the Hankel matrix generated by the given sequence $\{\bar{\mu}_{ij}\}_{i,j \in \mathcal{M}}$.

**Theorem 2.** *The sequence $\{\bar{\mu}_{ij}\}_{i,j \in \mathcal{M}}$ can determine the density functions $\bar{p}_N(x, y)$ if and only if it satisfies one of the following conditions:*
*(a) For any $m-1$-dimensional vectors $\bar{\mathbf{X}}_{m-1}^T = (x_1, \cdots, x_{m-1})$, $\mathbf{Y}_{m-1}^T = (y_1, \cdots, y_{m-1})$ and m-dimensional vectors $\mathbf{X}_m^T = (x_0, \cdots, x_{m-1})$, $\bar{\mathbf{Y}}_m^T = (y_1, y_2 \cdots, y_m)$ $(1 \le m \le M)$,*

$$
\bar{\mu}_{2m,0} \mathbf{Z}_{2m+1}^T \bar{\Gamma}_{2m+1} \mathbf{Z}_{2m+1} > (\bar{U}_{0.}^T \mathbf{X}_m + \bar{U}_{m.}^T \mathbf{Y}_{m-1})^2
$$

*and:*

$$
\bar{\mu}_{0,2m} \mathbf{Z}_{2m+1}^T \bar{\Gamma}_{2m+1} \mathbf{Z}_{2m+1} > (\bar{U}_{.m}^T \bar{\mathbf{X}}_{m-1} + \bar{U}_{0.}^T \bar{\mathbf{Y}}_m)^2;
$$

*(b)If there exists $m_0 \ge 1$ such that $\mathbf{Z}_{2m_0+1}^T \bar{\Gamma}_{2m_0+1} \mathbf{Z}_{2m_0+1} = 0$, then:*

$$
|\bar{\Gamma}_{2l+1}| = 0
$$

*for $m_0 \le l \le M$, where the vectors $\mathbf{Z}_{2m-1}^T = (X_m^T, x_m, Y_{m-1}^T, y_m)$ and $\bar{U}_{.0}^T = (\bar{\mu}_{m0}, \cdots, \bar{\mu}_{2m-1,0})$, $\bar{U}_{0.}^T = (\bar{\mu}_{0m}, \cdots, \bar{\mu}_{0,2m-1})$ and $\bar{U}_{.m}^T = (\bar{\mu}_{1m}, \cdots, \bar{\mu}_{m-1,m})$, $\bar{U}_{m.}^T = (\bar{\mu}_{m1}, \cdots, \bar{\mu}_{m,m-1})$.*

## 3. Maximum Entropy for Moment Problems

In Section 2, we discussed the existence of solutions of the moment problems. Now, we will use the maximum entropy method to get the solutions for Problems (6) and (7). Moreover, we will consider the convergence of maximum entropy density functions.

Given the first $N+1$ moments $\bar{\mu}_0, \bar{\mu}_1, \cdots, \bar{\mu}_N$, the core idea of the maximum entropy method is to find the probability density function $\bar{p}_N(x)$, such that:

$$
\bar{\mu}_j = \int_{\mathbb{R}} x^j \bar{p}_N(x) dx, \quad j = 0, 1, \cdots, N, \tag{23}
$$

where $\bar{\mu}_0 = 1$.

The Lagrange operator can be defined as:

$$
L(\bar{p}, \bar{\lambda}_0, \cdots, \bar{\lambda}_N) = - \int_{\mathbb{R}} \bar{p}(x) \ln \bar{p}(x) dx + \sum_{j=0}^{N} \bar{\lambda}_j \left( \int_{\mathbb{R}} x^j \bar{p}(x) dx - \bar{\mu}_j \right), \tag{24}
$$

where $\bar{\lambda}_j, j = 0, 1, \cdots, N$ are the Lagrange multipliers.

By the functional variation with respect to $\bar{p}(x)$, we have:

$$
\bar{p}_N(x) = \exp\left\{ -\sum_{j=0}^{N} \bar{\lambda}_j x^j \right\} = \max_{\bar{p}} \left\{ -\int_{\mathbb{R}} \bar{p}(x) \ln \bar{p}(x) dx \right\} \tag{25}
$$

and (23) holds. The values of $\bar{\lambda}_j$ can be calculated by solving a system of $N + 1$ equations resulting from the moments conditions (23). Here, we take $N = 2$ as an example and work out the values of $\bar{\lambda}_j$ $(j = 0, 1, 2)$ as:

$$\bar{\lambda}_0 = \bar{\mu}_1^2 + \ln \sqrt{2\pi(\bar{\mu}_2 - \bar{\mu}_1^2)}, \ \bar{\lambda}_1 = -\frac{\bar{\mu}_1}{\bar{\mu}_2 - \bar{\mu}_1^2}, \ \bar{\lambda}_2 = \frac{1}{2(\bar{\mu}_2 - \bar{\mu}_1^2)}.$$

So far, we have considered the existence of the solutions of the moment Problems (6) and (7) for all $N \geq 1$. According to Theorem 3.3.11 in Durrett [12], we know that the probability density function $\bar{p}(x)$ is also unique in the weak convergence (see (27)) as long as its moments $\{\bar{\mu}_j\}_{j=0}^{+\infty}$ satisfy the conditions (a) and (b) in Theorem 1 and:

$$\limsup_{m \to +\infty} \frac{\bar{\mu}_{2m}^{1/2m}}{2m} < \infty. \tag{26}$$

By the analysis in Frontini [13], we have the weak convergence for the maximum entropy solution $\bar{p}_N(x)$ as follows:

$$\lim_{N \to +\infty} \int_{\mathbb{R}} \bar{p}_N(x) \ln \bar{p}_N(x) dx = \int_{\mathbb{R}} \bar{p}(x) \ln \bar{p}(x) dx. \tag{27}$$

**Remark 2.** *By an argument analogous to the one-dimensional case, we can use the maximum entropy method to solve a concrete joint probability density function $\bar{p}_N(x, y)$ for the two-dimensional moment problem (20). The detailed process is shown in the next section and need not be repeated here.*

*By Theorem 2 and Theorem 14.20 in Schmüdgen [14], if the marginal moments $\{\bar{\mu}_{m\cdot}\}_{2m=0}^{+\infty}$ and $\{\bar{\mu}_{\cdot m}\}_{2m=0}^{+\infty}$ satisfy Conditions (a) and (b) in Theorem 2 and the following multivariate Carleman condition [14]:*

$$\sum_{m=0}^{+\infty} \bar{\mu}_{2m,i}^{-\frac{1}{2m}} = \sum_{m=0}^{+\infty} \bar{\mu}_{i,2m}^{-\frac{1}{2m}} = +\infty$$

*for $0 \leq i \leq N$, then there exists a unique joint probability density function $\bar{p}(x, y)$ satisfying (20).*

*The convergence rate for the maximum entropy density $\bar{p}_N(x, y)$ has been analyzed in Frontini [13], and the results are analogous to (27).*

## 4. Numerical Experiments

In this section, we conduct numerical experiments to calculate the two-dimensional maximum entropy density functions. In the numerical experiments, we use the maximum entropy method proposed in Section 3 to calculate the joint probability density functions for the weekly closing price of stock $X$ and the weekly rate of return $Y$ of Shanghai A shares.

By random sampling, we collect the data about the maximum (Table 1) and minimum (Table 2) values of the mixed sample moments of orders one and two for $X$ and $Y$, respectively. That is,

**Table 1.** The values for the maximum value of the moments.

| Moments | $\bar{\mu}_{00}$ | $\bar{\mu}_{10}$ | $\bar{\mu}_{20}$ | $\bar{\mu}_{11}$ | $\bar{\mu}_{01}$ | $\bar{\mu}_{02}$ |
|---------|------|--------|--------------------|--------|-------|--------|
| Values  | 1    | 30.781 | $1.5095 \times 10^3$ | 1.0158 | 0.033 | 0.0086 |

**Table 2.** The values for the minimum value of the moments.

| Moments | $\underline{\mu}_{00}$ | $\underline{\mu}_{10}$ | $\underline{\mu}_{20}$ | $\underline{\mu}_{11}$ | $\underline{\mu}_{01}$ | $\underline{\mu}_{02}$ |
|---------|------|--------|----------|---------|--------|--------|
| Values  | 1    | 14.311 | 343.1627 | $-1.2451$ | $-0.087$ | 0.0031 |

The two-dimensional maximum entropy problem is defined as: find a probability density function $\bar{p}_2(x,y)$ such that:

$$\bar{p}_2(x,y) = \max_{\bar{p}}\{-\int_{\mathbb{R}}\int_{\mathbb{R}}\bar{p}(x,y)\ln\bar{p}(x,y)dxdy\} \tag{28}$$

and:

$$\bar{\mu}_{kj} = \int_{\mathbb{R}}\int_{\mathbb{R}}x^k y^j \bar{p}_2(x,y)dxdy \tag{29}$$

for $k,j = 0,1,2$ and $0 \le k+j \le 2$.

Take notice of the formula as:

$$\bar{p}_2(x,y) = \bar{p}_2(y|x)\bar{p}_2(x), \tag{30}$$

where $\bar{p}_2(y|x)$ denotes the maximum entropy conditional density function. From (30), we can obtain the joint density function $\bar{p}_2(y|x)$ as long as we deduce $\bar{p}_2(y|x)$ and $\bar{p}_2(x)$. Let:

$$\bar{\mu}_{0j}(x) = \int_{R}y^j \bar{p}_2(y|x)dy, \quad j = 0,1,2, \tag{31}$$

be the conditional moments of the random variable $Y$ under $X$, which satisfy:

$$\int_{\mathbb{R}}x^k \bar{\mu}_{0j}\bar{p}_2(x)dx = \bar{\mu}_{kj} \tag{32}$$

for $k,j = 0,1,2$ and $0 \le k+j \le 2$.

According to Table 1 and the definition (22), we have:

$$\bar{\Gamma}_3 = \begin{pmatrix} 1 & 30.781 & 0.033 \\ 30.781 & 1509.5 & 1.0158 \\ 0.033 & 1.0158 & 0.0086 \end{pmatrix} \text{ and } \bar{\Delta}_{2\cdot} = \begin{pmatrix} 1 & 30.781 \\ 30.781 & 1509.5 \end{pmatrix}. \tag{33}$$

It is easy to verify that $\bar{\Gamma}_3$ and $\bar{\Delta}_{2\cdot}$ are positive definite. Hence, by Theorem 1, there exists a density function $\bar{p}_2(x)$ determined by $\{\bar{\mu}_{i0}, i = 0,1\}$.

To derive the explicit expressions of $\bar{p}_2(x,y)$, we need to construct the conditional moments $\{\bar{\mu}_{0j}(x)\}_{j=0}^{2}$ on the base of (32). Without loss of generality, we suppose $\bar{\mu}_{01}(x)$ is a constant. Combining (32) yields that:

$$\begin{cases} \bar{\mu}_{00}(x) = 1, \\ \bar{\mu}_{01}(x) = \bar{\mu}_{01}, \\ \bar{\mu}_{02}(x) = e^{-\bar{\delta}x^2} + \bar{\mu}_{01}^2, \quad \bar{\delta} = 40.7624. \end{cases} \tag{34}$$

By calculation, we derive the expressions of $\bar{p}_2(x,y)$ as follows:

$$\bar{p}_2(x,y) = \tfrac{1}{\sqrt{2\pi}}\exp\{-1 - \bar{\lambda}_0 - \bar{\lambda}_1 x - \bar{\lambda}_2 x^2 \\ -\tfrac{1}{2}\bar{\mu}_{01}^2 e^{-\bar{\delta}x^2} + \bar{\mu}_{01}ye^{-\bar{\delta}x^2} - \tfrac{1}{2}y^2 e^{-\bar{\delta}x^2}\}, \tag{35}$$

where $\bar{\lambda}_0 = 3.9276$, $\bar{\lambda}_1 = -0.0548$, and $\bar{\lambda}_2 = 0.0009$.

In the same way, with the data in Table 2, we can obtain the following probability density function $\underline{p}_2(x,y)$ determined by $\{\underline{\mu}_{k,j}, k,j = 0,1,2 \text{ and } 0 \le k+j \le 2\}$.

$$\underline{p}_2(x,y) = \frac{1}{\sqrt{2\pi}} \exp\{-1 - \underline{\lambda}_0 - \underline{\lambda}_1 x - \underline{\lambda}_2 x^2$$
$$-\tfrac{1}{2}\underline{\mu}_{01}^2 e^{-\underline{\delta}x^2} + \underline{\mu}_{01} y e^{-\underline{\delta}x^2} - \tfrac{1}{2}y^2 e^{-\underline{\delta}x^2}\}, \tag{36}$$

where $\underline{\lambda}_0 = 3.1240$, $\underline{\lambda}_1 = -0.1034$, $\underline{\lambda}_2 = 0.0036$, and $\underline{\delta} = 773.7203$.

Figures 1 and 2 show visually the sketches of the marginal entropy density functions $\bar{p}_2(x)$, $\underline{p}_2(x)$ (derived in (25) with $N = 2$) and the joint entropy density functions $\bar{p}_2(x,y)$, $\underline{p}_2(x,y)$ (derived in (35) and (36)), respectively.

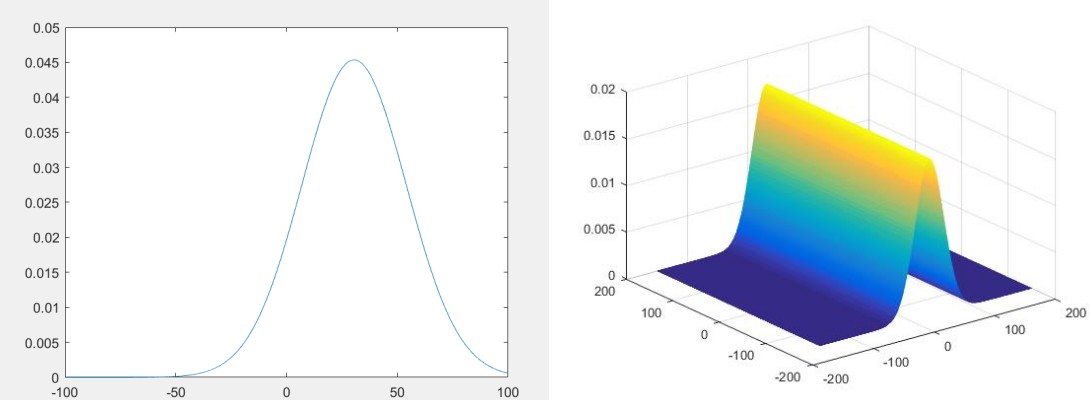

**Figure 1.** The figures above show the figures of the maximum entropy marginal density $\bar{p}_2(x)$ (**left**) and joint density $\bar{p}_2(x,y)$ (**right**) determined by $\{\bar{\mu}_{j0}\}_{j=0}^2$ and $\{\bar{\mu}_{ij}\}_{i,j\in\mathcal{M}}^2$, respectively.

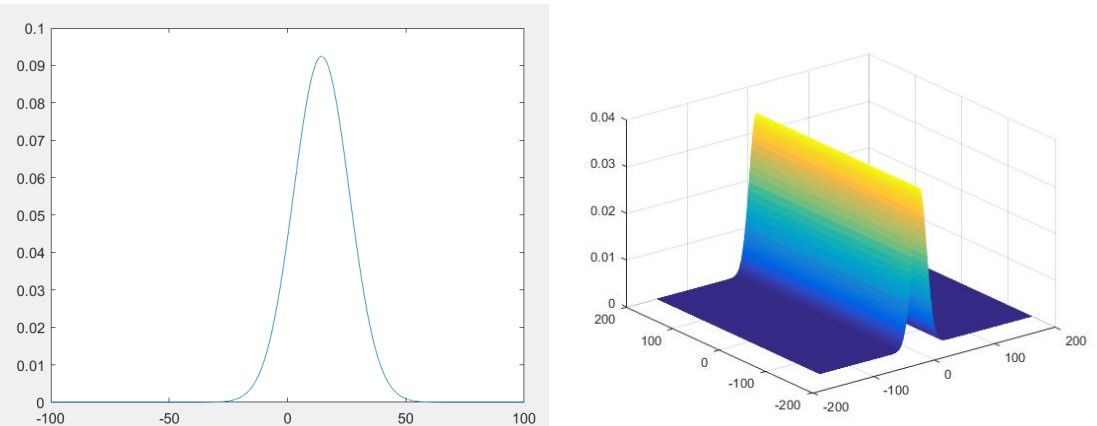

**Figure 2.** The figures above show the figures of the maximum entropy marginal density $\underline{p}_2(x)$ (**left**) and joint density $\underline{p}_2(x,y)$ (**right**) determined by $\{\underline{\mu}_{j0}\}_{j=0}^2$ and $\{\underline{\mu}_{ij}\}_{i,j\in\mathcal{M}}^2$, respectively.

**Author Contributions:** Conceptualization, L.G. and D.H.; methodology, D.H. and L.G.; software, L.G.; validation, L.G.; formal analysis, L.G.; investigation, L.G.; resources, L.G.; data curation, L.G.; writing—original draft preparation, L.G.; writing—review and editing, L.G.; visualization, L.G.; supervision, L.G. and D.H.; project administration, L.G.; funding acquisition, D.H.

**Funding:** This work is supported by National Natural Science Foundations of China [grant number 11531001] and National Program on Key Basic Research Project [grant number 2015CB856004].

**Conflicts of Interest:** The authors declare no conflict of interest.

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
