# Peer review of "Methods of Moment and Maximum Entropy for Solving Nonlinear Expectation"

_mathematics, doi:10.3390/math7010045_

Round 1

Reviewer 1 Report

The paper by Gao and Han discusses the methods of moments and maximum entropy for
solving nonlinear expectation. The approach seems fine. My main question is if this approach is applied to a master equation then is it the same as presented in this paper: Smadbeck, Patrick, and Yiannis N. Kaznessis. "A closure scheme for chemical master equations." Proceedings of the National Academy of Sciences 110.35 (2013): 14261-14265. The authors should explain the differences and the novelty of their paper more carefully.

Reviewer 2 Report

The manuscript is difficult to read because of its poor English. Also, it can benefit from adding more explanation to the introduction on the benefit of their proposed approach to solve nonlinear expectation with respect to other existing methods.

This is more evident in their numerical experiments, in which a proper definition of the problem and objective and what they are computing using their theoretical method and what they are comparing with the data is not very clear. Also, the figures, which supposed to show the density functions, are not clearly explained. What the figures supposed to show and what we supposed to understand from this numerical evaluation? Why the univariate density is 2d and the bivariate density kind of singular? More explanation on the numerical experiment and how this supports their method can be very useful for the reader to grasp the main message and power of their method.
